# Reporting Standards for a Bland–Altman Agreement Analysis: A Review of Methodological Reviews

**DOI:** 10.3390/diagnostics10050334

**Published:** 2020-05-22

**Authors:** Oke Gerke

**Affiliations:** 1Department of Nuclear Medicine, Odense University Hospital, Kløvervænget 47, 5000 Odense, Denmark; oke.gerke@rsyd.dk; 2Department of Clinical Research, University of Southern Denmark, 5000 Odense, Denmark

**Keywords:** agreement, Bland–Altman plot, confidence interval, interrater, Limits of Agreement, method comparison, repeatability, reporting, reproducibility, Tukey mean-difference plot

## Abstract

The Bland–Altman Limits of Agreement is a popular and widespread means of analyzing the agreement of two methods, instruments, or raters in quantitative outcomes. An agreement analysis could be reported as a stand-alone research article but it is more often conducted as a minor quality assurance project in a subgroup of patients, as a part of a larger diagnostic accuracy study, clinical trial, or epidemiological survey. Consequently, such an analysis is often limited to brief descriptions in the main report. Therefore, in several medical fields, it has been recommended to report specific items related to the Bland–Altman analysis. The present study aimed to identify the most comprehensive and appropriate list of items for such an analysis. Seven proposals were identified from a MEDLINE/PubMed search, three of which were derived by reviewing anesthesia journals. Broad consensus was seen for the a priori establishment of acceptability benchmarks, estimation of repeatability of measurements, description of the data structure, visual assessment of the normality and homogeneity assumption, and plotting and numerically reporting both bias and the Bland–Altman Limits of Agreement, including respective 95% confidence intervals. Abu-Arafeh et al. provided the most comprehensive and prudent list, identifying 13 key items for reporting (Br. J. Anaesth. 2016, 117, 569–575). An exemplification with interrater data from a local study accentuated the straightforwardness of transparent reporting of the Bland–Altman analysis. The 13 key items should be applied by researchers, journal editors, and reviewers in the future, to increase the quality of reporting Bland–Altman agreement analyses.

## 1. Introduction

The Bland–Altman Limits of Agreement (BA LoA), or simply Bland–Altman plots, are used widely in method comparison studies with quantitative outcomes, as evidenced by more than 34,325 citations of the seminal *Lancet* paper to date [1]. In this analysis, a pair of observations is made from the same subject, with two different methods. Subsequently, the means and differences of these pairs of values for each subject are displayed in a scatter plot. The plot usually also shows a line for the estimated mean difference between the two methods (a measure of the bias between the two methods), and lines indicating the BA LoA (within which approximately 95% of all population differences would lie) [1,2,3,4]. Use of the BA LoA assumes that the differences are normally distributed (e.g., [5]). 

The BA LoA are sample-based estimates of the true, but unknown population limits. Bland and Altman [1] originally proposed approximate confidence intervals to assess the precision of these estimates. Over the last decades, the importance of confidence intervals in general [6,7], and for the BA LoA in particular [8,9,10], was emphasized. Carkeet [11] proposed exact confidence intervals for the BA LoA using two-sided tolerance factors for a normal distribution. Zou [12] and Olofsen et al. [13] provided confidence intervals for the BA LoA in case of multiple paired observations in each subject. The latter extended their work by offering a freely-available online implementation, accompanied by a formal description of the more advanced Bland–Altman comparison methods [14], and they proposed a standard format of reporting a BA agreement analysis. However, recent examples of Bland–Altman plots employing samples as small as 31 [15], 6 and 12 [16], or 14 [17] can be found easily in MEDLINE/PubMed (by searching for “Bland–Altman” and “Agreement” and limiting the search to publication dates in August 2019). These studies did not quantify the uncertainty of the estimated BA LoA by means of confidence intervals. Vock [18] emphasized that a tolerance interval or the outer confidence limits for the BA LoA alone can also provide a range that will contain at least a specified percentage of the future differences with a known certainty. Kottner et al. [19] pointed out that agreement and reliability assessment is either conducted in dedicated studies with a respective primary focus or as a part of larger diagnostic accuracy studies, clinical trials, or epidemiological surveys that report agreement and reliability as a quality control. The latter is often done in subsamples, resulting in small to moderate sample sizes [19]. 

The *Guidelines for Reporting Reliability and Agreement Studies* (GRRAS) by Kottner et al. [19] comprise a comprehensive checklist of 15 items that support the transparent reporting of agreement and reliability studies (see Table 1 in [19]). Item no. 10 and 13 relate to the description of the statistical analysis and reporting of estimates of reliability and agreement, including measures of statistical uncertainty. More recently, these aspects have been commented upon in more general terms [20]. Stöckl et al. [21] proposed adding pre-defined error limits for both the outer confidence limits of the BA LoA and the confidence limits of the estimated bias. Additionally, researchers have attempted to establish reporting standards for BA plots in various fields (e.g., [22,23]). 

The present study aimed to identify reporting standards for BA agreement analyses, to single out the most comprehensive and appropriate list, and to exemplify this proposal using data from an interrater Bland–Altman analysis conducted at our institution.

## 2. Materials and Methods 

### 2.1. Literature Review and Data Extraction

MEDLINE/PubMed was searched for (((*reporting*) *OR checklist*)) *AND* ((((*“method comparison”*) *OR bland–altman*) *OR* (*bland AND altman*)) *OR agreement*) within the timeframe of 1 January 1983 to 3 March 2020 and without further field restrictions (see Appendix A). The list of article summaries that emerged from the search was screened, and potentially relevant titles were chosen for a full-text review. All papers that proposed a list of items for standardizing BA plots were included, irrespective of article type (systematic reviews, narrative reviews, original articles, commentaries, editorials, letters, and case studies). The *Preferred Reporting Items for Systematic Reviews and Meta-Analysis* (PRISMA) statement [24] was consulted but it could only be applied to a limited extent due to the methodological nature of papers to be included in this review. 

Included articles were described by their core characteristics (first author, year of publication, field/area, target journals and time frame in case of inherent reviews, and evidence base (e.g., number of included papers)). Subsequently, lists of proposed items were compared, and the most comprehensive list was chosen and applied to the following worked example.

### 2.2. Worked Example

In a recent study on the association of left atrium enlargement (measured by non-contrast computed tomography) with traditional cardiovascular risk factors, intra- and interrater differences were analyzed using the BA LoA, supplemented by their exact 95% confidence intervals according to Carkeet [11], and shown as Supplemental Figure [25]. Raw data have been provided in Appendix A.

## 3. Results

### 3.1. Reporting Standards for Bland–Altman (BA) Agreement Analysis

The literature search resulted in 5551 hits. This is likely to be sensitive but not very specific as search results comprised as farfetched topics as negotiation of protective sex in West African countries, understanding foreign accent syndrome, effect of playing position on injury risk in male soccer players, and agreement in reporting between trial publications and current clinical trial registry in high impact journals [26,27,28,29]. After screening 5551 titles, 28 full-text articles were assessed (Figure 1). Six publications were identified by the database search [13,22,23,30,31,32], and one further article was found as a reference in the aforementioned publications [33]. 

Three out of seven studies were published in anesthesia journals, while the remaining stemmed from various fields (Table 1). Four studies were based on publications within 2-year time frames, and the number of included publications varied from 0 to 394 (median: 50). 

Sixteen reporting items were proposed across the included studies. Broad consensus was seen for the a priori establishment of acceptable LoA (Item #1, Table 2); estimation of repeatability of measurements in case of available replicates within subjects (#3); visual assessment of a normal distribution of differences and homogeneity of variances across the measurement range (#4); and plotting and numerically reporting both bias and the BA LoA, including respective 95% confidence intervals (#6–9). A description of the data structure (#2), between- and within-subject variance (or stating that confidence intervals for the BA LoA were derived by accounting for the inherent data structure; #11), and distributional assumptions (#13) followed. Only one review raised the issue of a sufficiently wide measurement range (#10), sample size determination (#14), or correct representation of the *x*-axis (#15). Upfront declaration of conflicts of interest (#16) also appeared only once, but this can generally be presumed to be covered by the ethics of authorship.

The list of reporting items proposed by Abu-Arafeh et al. [23] was the most comprehensive (13 out of 16 items), followed by those proposed by Montenij et al. [31] (10 out of 16 items) and Olofsen et al. [13] (9 out of 16 items). The latter two lists were complete subsets of Abu-Arafeh et al.’s [23] list, with the exception of Item #14 on the list by Montenij et al. [31]. The most recently published list by Flegal et al. [30] comprised items that were derived as a modified version of those suggested by Abu-Arafeh et al. [23]. Specifically, they omitted items related to statistical software and repeated measurements, as the latter are rarely applied in studies entailing self-reported weight and height [30].

Abu-Arafeh et al. [23] identified 111 papers that were potentially relevant to the question “what do authors recommend should be reported when a Bland and Altman analysis is presented?”. This material was used to collect suggestions for potential items, and the most frequent and relevant ones were consolidated (Supplementary Material B in [23]). Nine out of 13 key items were selected as most popular features on the consolidated list (#1–9). The remaining 4 key items were chosen because
a guidance paper clearly showed the dependence of the usefulness of the BA LoA on the range of the values studied (#10);replicated data affects the calculation of confidence intervals for the BA LoA (#11); anddetails on computing methods are desirable in any report using statistics (#12, #13).

Across guideline proposals, there seems to be a tacit consensus of the fact that the *x*-axis must show average values of the two methods compared (#15), as also discussed by Bland and Altman [35]. The issue of sample size determination (#14) will be discussed later.

### 3.2. Worked Example

The list of reporting items proposed by Abu-Arafeh et al. [23] was applied in a clinical setting [25], using item numbers from Table 2 as subsections.

#### 3.2.1. Pre-Establishment of Acceptable Limits of Agreement (LoA)

Unfortunately, acceptable LoA were not established a priori. From the clinical perspective, the judgement of whether the BA LoA of −2.39 to 2.36 with outer 95% confidence limits −2.71 and 2.69 were sufficiently close had to be made post hoc, in light of a mean area of 24.1 cm^2^ (range: 12.9–44.3). 

#### 3.2.2. Description of the Data Structure

Left atrium size was measured by 7 trained radiographers (raters 1–7 in Appendix A), blinded to all clinical data. Each radiographer repeated left atrium area measurements in 20 participants. A second reader (Dr. Maise Høigaard Fredgart) measured left atrium size once in the 140 participants included in the original study.

#### 3.2.3. Estimation of Repeatability

Repeated measurements were performed only for the 7 radiographers. Mean and standard deviations of these intrarater differences were 0.59 and 1.64, respectively. 

Repeatability coefficients were derived by means of a linear mixed effects model, with *repetition* as fixed and *patient* and *rater* as random factors [36]. The repeatability coefficient for a new rating of the same scan by the same rater equaled 2.77 times √1.019588 = 2.80 (Appendix A). The repeatability coefficient for a new rating of the same scan by *any rater* equaled 2.77 times √(1.019588+0.1400625) = 2.98.

#### 3.2.4. Plot of the Data and Visual Inspection for Normality and Absence of Trend

A histogram of the differences suggested a roughly normal distribution (Figure 2). The correlation between differences and means was 0.08. The null hypothesis of equal means and variances, tested using the Bradley–Blackwood procedure [37], could not be rejected (*p* = 0.64). The scatter of differences did not depend on the means, indicating homogeneity of variances across the measurement range (Figure 3). 

#### 3.2.5. Transformation of the Data

Not applicable as data were judged to be roughly normally distributed (Figure 2).

#### 3.2.6. Plotting and Numerically Reporting the Mean of the Differences (Bias)

The estimated bias was –0.01 (Figure 3).

#### 3.2.7. Estimation of the Precision of the Bias

The 95% confidence interval for the mean difference (with unknown population standard deviation) was –0.21 to 0.19. This area has been highlighted in olive-teal in Figure 3.

#### 3.2.8. Plotting and Numerically Reporting the BA LoA

The BA LoA were –2.39 and 2.36 (Figure 3). 

#### 3.2.9. Estimation of the Precision of the BA LoA

The 95% confidence intervals for the BA LoA were –2.71 to –2.14, and 2.12 to 2.69. These areas have been highlighted in light blue in Figure 3.

#### 3.2.10. Indication of Whether the Measurement Range Is Sufficiently Wide

According to the Preiss–Fisher procedure [34], the measurement range was sufficiently wide. The interrater data was randomly mispaired 1000 times, and the distribution of the respective 1000 standard deviations was clearly beyond the observed standard deviation of the differences (1.21), with a minimum of 7.12 (Figure 4). 

#### 3.2.11. Between- and within-Subject Variance or Stating that the Confidence Intervals of the BA LoA Were Derived by Taking the Data Structure into Account 

The confidence intervals of the BA LoA were derived by taking the data structure into account. Non-repeated, paired ratings were used as replicates were only performed for the 7 radiographers. However, the inclusion of repeated measurements for the main investigator would have yielded two paired observations per subject. This would reflect the inherent natural variation within the rater, and thereby provide a more precise picture of agreement [4].

#### 3.2.12. Software Package or Computing Processes Used

All analyses were conducted using STATA/MP 16.1 (StataCorp, 4905 Lakeway Drive, College Station, TX 77,845, USA).

#### 3.2.13. Distributional Assumptions Made

The differences were assumed to be normally distributed (see Figure 2 above).

## 4. Discussion

### 4.1. Statement of Principal Findings

The present study revealed 7 lists of reporting items for BA analysis, with most stemming from the field of anesthesiology. The work of Abu-Arafeh et al. [23] turned out to be the most comprehensive and prudent, identifying 13 key items. The work of Olofsen et al. [13,14] stood out due to its detailed discussion of confidence intervals in case of multiple paired observations and its concreteness with respect to implementation. 

### 4.2. Strengths and Weaknesses of the Study

Research traditions vary across fields with respect to method comparison studies on quantitative outcomes, in particular when contrasting clinical and laboratory research [21,38,39]. This review compared proposals for unified reporting of BA plots from diverse medical areas to obtain suggestions for items that should be consulted and complied with in BA agreement analyses in future [23].

The main limitations of this work are the ad hoc literature search in only one database (MEDLINE/PubMed) and the fact that only one reviewer screened titles and abstracts, read full-texts, extracted data, and performed the qualitative evidence synthesis [24]. This may have led to ignoring other lists of essential reporting items than those identified. However, those identified appear to be reasonable and comprehensive from our point of view. Another limitation is that the analysis of the worked example [25] with a Bland–Altman analysis oversimplified the underlying data structure because one physician was compared to a set of radiographers. The BA LoA were thereby underestimated slightly as compared to the repeatability coefficients derived in Section 3.2.3.

### 4.3. Meaning of the Study: Possible Mechanisms and Implications for Clinicians or Policymakers

With respect to the statistical analysis of agreement with BA plots, the following three reasons obligate researchers to employ more diligence and warrant journal editors to consider the work of Abu-Arafeh et al. [23] as an essential supplement to GRRAS [19]:the unbowed popularity of BA plots, as judged by the number of citations of the seminal papers [1,3] (and a pedagogical update [4]);the limited extent to which BA plots are reported and interpreted in the literature (e.g., [15,16,17,40]); andthe associated need for standardization, as evidently expressed by the included proposals [13,22,23,30,31,32,33].

### 4.4. Unanswered Questions and Future Research

#### 4.4.1. Sample Size Considerations

Only Montenij et al. [31] incorporated the motivation of sample sizes in their proposed list. Bland and Altman [1,4,41] considered the study of agreement an estimation issue and, therefore, they proposed to base the sample size on the width of the approximate 95% confidence intervals for the LoA. Others have reinforced this approach [8,42,43,44]. Donner and Zou [45] proposed closed-form confidence intervals for the BA LoA, based on the Method of Variance Estimates Recovery (MOVER). Carkeet [11] suggested exact two-sided tolerance intervals for the BA LoA, which can be computed easily using the Supplementary Digital Content accompanying his article. Carkeet and Goh [10] and Shieh [46] compared approximate and exact confidence intervals for the LoA. Moreover, Shieh [46] and Jan and Shieh [47] provided SAS and R codes for precise interval estimation and sample size derivation such that (a) the expected confidence interval width is within a given threshold and (b) this width will not exceed an a priori defined value with a given assurance probability. Prior to such recommendation, the rule-of-thumb of “50 subjects with three replicate measurements on each method” [48,49] or “at least 100 subjects” [43] was suggested.

Another perspective on agreement assessment considers it in a testing framework. Lin et al. [50] proposed a two one-sided tests (TOST) procedure employing the approximate confidence intervals recommended by Bland and Altman [1,4,41]. Choudhary and Nagaraju [51] modified a nearly unbiased test for the assessment of agreement using the probability criteria recommended by Wang and Hwang [52]. These have led to a simple closed form approximation that is numerically equivalent to the exact test for n < 30 [51]. Yi et al. [53] extended the repeatability coefficient approach and proposed an equivalence test for agreement between two or more measurement tools or raters. Liao [54] determined sample size using an interval approach for concordance, which is based on the discordance rate and a tolerance probability. In contrast to Lin et al. [50], Lu et al. [55] proposed a sample size rationale that allows for bias between two methods. Further, their method considered the type II error probability in addition to the type I error probability, mean and the standard deviation of the paired differences, and predefined lower and upper limits. Shieh [56] compared several TOST procedures for assessing agreement between two methods, and described an improved and exact approach for declaring agreement. SAS and R codes were presented for conducting the equivalence test for agreement, power calculation, and sample size determination.

In the context of method comparison studies, though TOST procedures date back to 1998 [50], software implementations have been offered only recently [56]. During the past 34 years, attention has continuously been drawn to the precision of the BA LoA in terms of approximate confidence interval width [1,4,8,41,42,43,44,45]. However, lately, the focus has shifted to the width of exact confidence intervals for the BA LoA [11,46,47]. Possible reasons for a larger focus on estimation rather than testing approaches may be that:due to cost, time, and practical restrictions, a large proportion of agreement studies are conducted only for quality assurance in (comparably small) subgroups of subjects of the main investigation; e.g., data for our worked example stemmed from an epidemiological study on 14,985 subjects;in a study with repeated measurements, it is easier to conduct sample size planning in terms of confidence interval width rather than using a TOST procedure [12,13,56]; andthe a priori definition of acceptable limits for the BA LoA may be more challenging in imaging studies (comparing raters) than it is in method comparison studies (comparing methods or instruments).

#### 4.4.2. Insufficient Vigor of Statistical Advice

Since Altman [57] indicated that “huge sums of money are spent annually on research that is seriously flawed through the use of inappropriate designs, unrepresentative samples, small samples, incorrect methods of analysis, and faulty interpretation … much poor research arises because researchers feel compelled for career reasons to carry out research that they are ill equipped to perform, and nobody stops them,” reporting guidelines such as the *Standards for Reporting Diagnostic Accuracy* (STARD) and PRISMA aim to improve the quality of reported research [58]. Unfortunately, these guidelines are adopted and adhered to much lesser than they should be [59]. Moreover, in 2014, Smith [60] reckoned that Altman’s editorial [57] could be republished almost unchanged because quality assurance methods, such as better equipping ethics committees to detect scientific weaknesses and more journals employing statistical reviewers, did not prevent published literature from continuing to be largely misleading and of low quality. By contrast, Bhopal [61] postulated that neither the quantity of research conducted nor the motivation behind it is relevant. He added that more good research will be conducted if the work of the novice and the enthusiast is encouraged, and if talent is spotted and developed at the price of more supervision and close peer review. However, unsatisfactory work will be published occasionally when supervision and peer review fail [61]. 

Why are guidelines and statistical advice not followed optimally? The pressure on individual researchers continues to reflect the pressure on institutions, which are judged by the crude number of research publications and on grant income [62]. In Denmark, the Ministry of Education and Research awards points to publications in indexed journals, which, in turn, are maintained by area-specific experts [63]. Annually accumulated points are then compared across research institutions, which influences the distribution of base funds. Furthermore, grant income has become the main financial source for the creation and maintenance of PhD and postdoc positions. The documentation of prior subject-related research is undoubtedly one important requirement for a successful funding application. Reed [64] postulated that the production of research funding applications has turned into assembly-line work because, considering the fierce competition, chances of obtaining funding are often about 10% or less. In clinical development, pharmaceutical companies have to demonstrate to authorities (like the US American Food and Drug Administration) that all members of clinical trial teams (e.g., study manager, clinical research associate, data manager, statistician, medical writer) are trained suitably. In contrast, in academic research, there was a serious shortage of statisticians to teach and, especially, to participate in research in 1990 [65], and this is very likely still true today [66]. Lack of training during education and sparring in research teams with a statistician is, then, accompanied by in-depth statistical refereeing only in major journals. 

Even explicit editorial guidance on improved statistical reporting practice may be insufficient. Curran-Everett and Benos [67] published 10 guidelines for reporting statistics in journals published by the American Physiological Society. They reviewed 10% of articles published in target journals the year before and the second year after the guidelines were proposed [68]. They concluded that the guidelines did not have an effect on the occurrence of standard errors, standard deviations, confidence intervals, and precise *p*-values. This supports their original assertion that authors, reviewers, and editors must make a joint effort in using and reporting statistics consistent with best practices. Furthermore, the *Journal of Physiology* and the *British Journal of Pharmacology* jointly undertook a similar editorial series in 2011 [69,70,71]. Subsequently, Diong et al. [72] reviewed randomly-sampled research papers published in these journals before and after the publication of the editorials. They found no evidence of improvement in reporting practices following the publication of the editorial advice. Therefore, they concluded that stronger incentives, better education, and widespread enforcement are needed for creating enduring improvements in reporting practices. 

Undoubtedly, training in research methodology in general, and especially in statistics, is the key to well-conducted research. In pre-graduate education, the MSc thesis is usually the first account of a research project. At the University of Southern Denmark, the MSc education in medicine was reformed in 2019. It now offers 5 profiles, one of which focusses on research. In postgraduate education, PhD programs are dedicated to research education. Otherwise, statistical guidance and support needs to be acquired at one’s research institution or within one’s own research network. Ultimately, it is evident that a research paper will live up to best practices in the use of statistical methods and respective reporting of results only if authors, reviewers, and editors make a joint effort [68]. Authors’ adherence to reporting guidelines will be supported by dedicated supervision [61], but it can only be enforced by adequate journal policies and critical peer-review. 

#### 4.4.3. Future Research

Future studies need to explore nonparametric alternatives to the BA LoA in case of non-normality of the data when normalizing transformations fail [1,4,42]. However, in doing so, they need to go beyond simple empirical estimates for 2.5% and 97.5% percentiles [4].

Classically, a BA analysis concerns the study of method comparison, i.e., the comparison of two methods or instruments. The agreement of more than two methods, instruments, or, especially, raters is also an area for further research [73,74,75].

Further, assuming that a basic set-up employs two raters performing two measurements on each scan or subject, it is essential to consider what advice can be given in multi-rater situations with respect to the number of raters, number of replicates, and sampling procedure [20,76]. 

## 5. Conclusions

Considering GRRAS as a broad reporting framework for agreement and reliability studies, Abu-Arafeh et al. [23] concretized its Item 10 (statistical analysis) and 13 (estimates of reliability and agreement including measures of statistical uncertainty) in the context of the Bland–Altman analysis in method comparison studies. A rigorous application of and compliance with the 13 key items recommended by Abu-Arafeh et al. [23] would increase both the transparency and quality of such agreement analyses. 

## Figures and Tables

**Figure 1 diagnostics-10-00334-f001:**
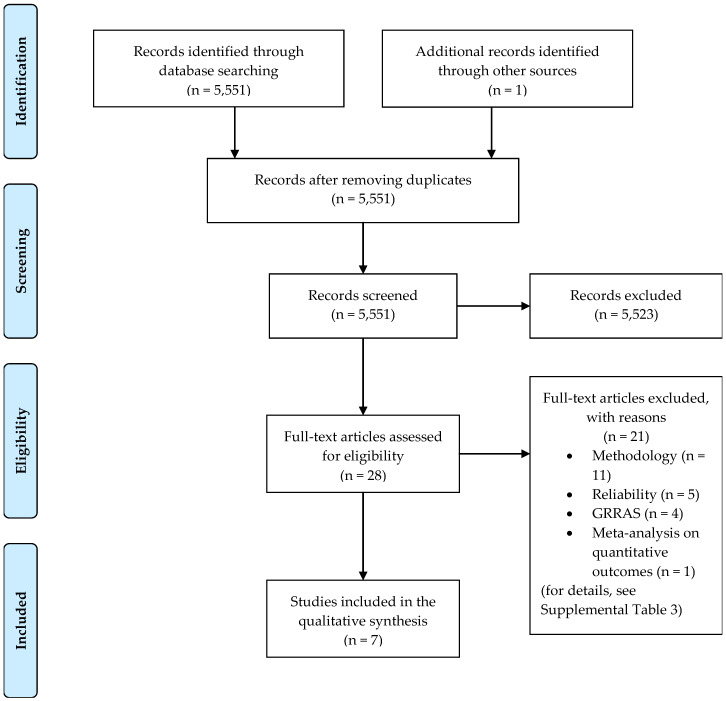
Preferred Reporting Items for Systematic Reviews and Meta-Analysis (PRISMA) flow diagram. GRRAS: Guidelines for Reporting Reliability and Agreement Studies [19].

**Figure 2 diagnostics-10-00334-f002:**
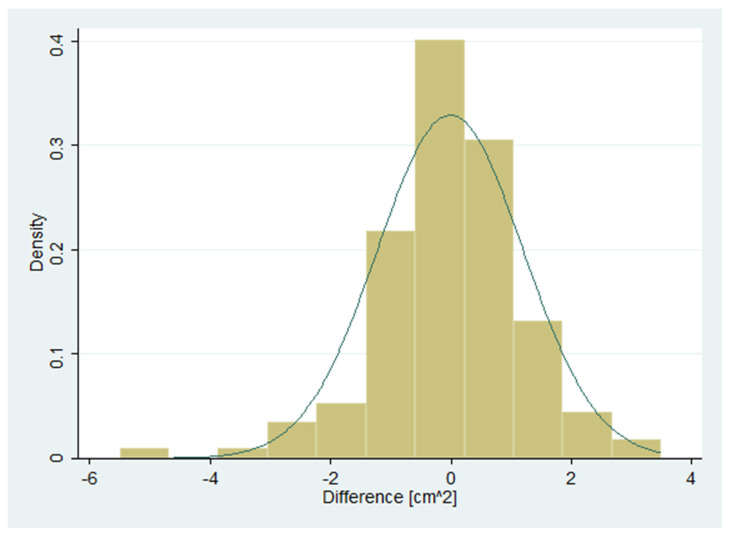
Histogram with approximating normal distribution of interrater differences (n = 140).

**Figure 3 diagnostics-10-00334-f003:**
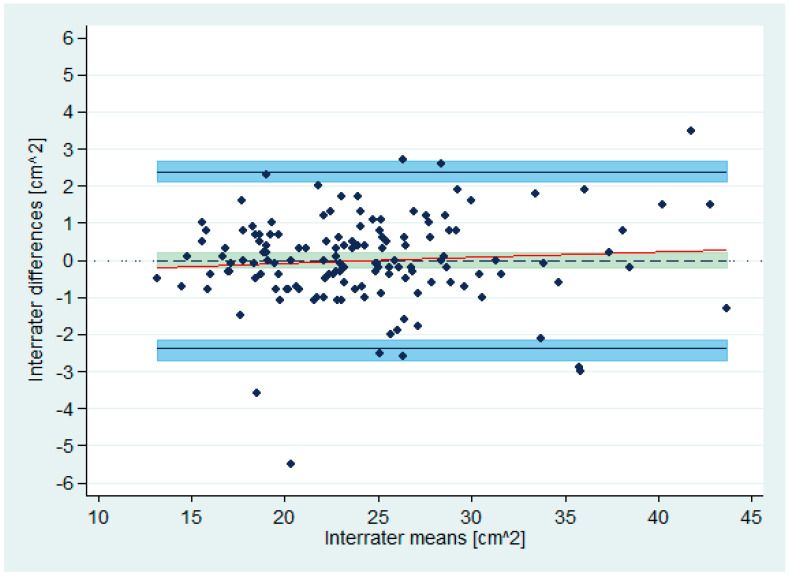
Bland–Altman plot for interrater agreement analysis (n = 140). Limits of Agreement are shown as solid, black lines with 95% confidence intervals (light blue areas), bias (as dotted black line) with 95% confidence interval (olive-teal area), and regression fit of the differences on the means (as solid red line).

**Figure 4 diagnostics-10-00334-f004:**
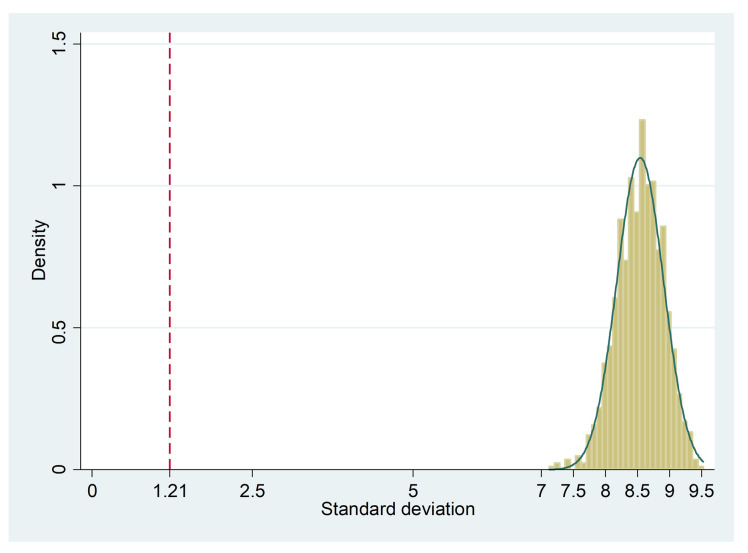
Distribution of the standard deviation from 1000 random mispairings according to the Preiss-Fisher procedure [34]. The observed standard deviation in the interrater sample (1.21) is clearly smaller than the minimum of the standard deviations from 1000 random mispairings (7.12).

**Table 1 diagnostics-10-00334-t001:** Characteristics of studies included.

Publication	Field/Area	Search Approach or Target Journals	Time Frame	Evidence Base
Flegal (2019) [30]	Self-reported vs. measured weight and height	Unrestricted; reference lists of systematic reviews, repetition of 2 PubMed searches of these, “related articles” in PubMed	1986–May 2019	N = 394 published articles
Abu-Arafeh (2016) [23]	Anesthesiology	Anaesthesia, Anesthesiology, Anesthesia & Analgesia, British Journal of Anaesthesia, Canadian Journal of Anesthesia	2013–2014	N = 111 papers
Montenij (2016) [31]	Cardiac output monitors	N/A	N/A	Expert opinion
Olofsen (2015) [13]	Unrestricted	N/A	N/A	Narrative literature review and Monte Carlo simulations
Chhapola (2015) [22]	Laboratory analytes	PubMed and Google Scholar	2012 and later	N = 50 clinical studies
Berthelsen (2006) [33]	Anesthesiology	Acta Anaesthesiologica Scandinavica	1989–2005	N = 50
Mantha (2000) [32]	Anesthesiology	Seven anesthesia journals	1996–1998	N = 44

N/A: not applicable.

**Table 2 diagnostics-10-00334-t002:** Comparison of proposed reporting items for a Bland–Altman analysis across included studies.

Reporting Item	Flegal (2019) [30]	Abu-Arafeh (2017) [23]	Montenij (2016) [31]	Olofsen (2015) [13]	Chhapola (2015) [22]	Berthelsen (2006) [33]	Mantha (2000) [32]
(1) Pre-established acceptable limit of agreement	X	X	X		X	X	X
(2) Description of the data structure (e.g., no. of raters, replicates, block design)		X	X	X			
(3) Estimation of repeatability of measurements if possible (mean of differences between replicates and respective standard deviations)		X		X	X	X	X
(4) Plot of the data, and visual inspection for normality, absence of trend, and constant variance across the measurement range (e.g., histogram, scatter plot)	X	X	X	X	X	X	X
(5) Transformation of the data (e.g., ratio, log) according to 4), if necessary		X				X	
(6) Plotting and numerically reporting the mean of the differences (bias)	X	X	X	X			
(7) Estimation of the precision, i.e., standard deviation of the differences or 95% confidence interval for the mean difference	X	X	X	X		X	
(8) Plotting and numerically reporting the BA LoA	X	X	X	X	X		
(9) Estimation of the precision of the BA LoA by means of 95% confidence intervals	X	X	X	X	X	X	X
(10) Indication of whether the measurement range is sufficiently wide (e.g., apply the Preiss-Fisher procedure [34])		X					
(11) Between- and within-subject variance or stating that the confidence intervals of the BA LoA were derived by taking the data structure into account		X	X	X			
(12) Software package or computing processes used		X		X			
(13) Distributional assumptions made (e.g., normal distribution of the differences)	X	X	X				
(14) Sample size considerations			X				
(15) Correct representation of the *x*-axis					X		X
(16) Upfront declaration of conflicts of interest						X	

BA LoA: Bland–Altman Limits of Agreement.

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
