# Peer review of "Reporting Standards for a Bland–Altman Agreement Analysis: A Review of Methodological Reviews"

_diagnostics, 2020, doi:10.3390/diagnostics10050334_

Round 1
Reviewer 1 Report
The current study summarizes seven previously published articles to identify the key items to report Bland-Altman Limits of Agreement (BA LoA). In addition, the author also reported 13 key items of BA LoA from a local study as an example. Here are reviewer comments and thoughts regarding the article:
Since this is a review article, more explanation of how articles were searched and screened,should be included. For example, how many articles were yielded when searched with the mentioned keywords; then how authors narrow down to seven articles? What are the specific criteria to short down the list? The author mentions only single sentence as “All papers that proposed a list of items for standardizing BA plots were included”; more explanation is needed.
Out of seven papers, the earliest paper was from year 2000. Was existing literature search vigorously or what were specific criteria?
Author chose 13 items to report BA LoA as suggested by Abu-Arafeh et al. Reviewer is curious why items from 14-16 from Table 2 were missing? Items 14-16 have been included in other papers to express BA LoA. What will be the rationale to omit item 14 (sample size considerations)?
Reviewer is a little confused regarding the nature of this article. At first, reviewer thought it’s a review paper, but then reviewer found that some data analysis for BA LoA was also presented. If it’s a review paper, more thorough investigation for BA LoA representation and those seven articles were needed. As a review paper, more details are needed. What are the
strength and lacking points of those articles? What are the author thoughts on current key points of expressing BA LoA? What is missing and what could be added in future? As a review paper, audience expectation will more in terms of the details of standard method of reporting BA LoA.
The reviewer thought it can be better to divide this article into two separate articles. Right now, the current status of the manuscript is neither a full review article nor an original research article. One as a full review paper where more details to present. Second can be an application paper where author can utilize his data to determine the items to present BA LoA with more details of each item.
Author Response
Dear Reviewer 1.
Thank you very much for your comprehensive and helpful comments. In the following, all items raised are addressed point-by-point.
Best wishes, Oke Gerke
The current study summarizes seven previously published articles to identify the key items to report Bland-Altman Limits of Agreement (BA LoA). In addition, the author also reported 13 key items of BA LoA from a local study as an example. Here are reviewer comments and thoughts regarding the article:
Since this is a review article, more explanation of how articles were searched and screened, should be included. For example, how many articles were yielded when searched with the mentioned keywords; then how authors narrow down to seven articles? What are the specific criteria to short down the list? The author mentions only single sentence as “All papers that proposed a list of items for standardizing BA plots were included”; more explanation is needed. Out of seven papers, the earliest paper was from year 2000. Was existing literature search vigorously or what were specific criteria?
Answer: In Section 2.1, the following details have been added. A PRISMA flowchart (new Figure 1) shows how 5,551 hits were narrowed down to 7 hits, and details of the search have been added as new Suppl. Data 1. The time frame was restricted to Jan 01, 1983 until March 03, 2020. The start date was motivated by the date of the very first publication of the Bland-Altman approach in The Statistician (1983;32(3):307-317; https://doi.org/10.2307/2987937). The list was actually solely reduced by looking for concrete lists of reporting items. In Section 4.2, the limitations of the study, especially with respect to the search itself, have been extended.
Author chose 13 items to report BA LoA as suggested by Abu-Arafeh et al. Reviewer is curious why items from 14-16 from Table 2 were missing? Items 14-16 have been included in other papers to express BA LoA. What will be the rationale to omit item 14 (sample size considerations)?
Answer: In Section 3.1, reasons for neglecting items 15 and 16 have been added, and the reader is referred to the Discussion section regarding item 14 (sample size). In Section 4.4.1, sample size determination in agreement studies is now discussed in more detail, and possible reasons for this item’s absence on most checklists are given.
Reviewer is a little confused regarding the nature of this article. At first, reviewer thought it’s a review paper, but then reviewer found that some data analysis for BA LoA was also presented. If it’s a review paper, more thorough investigation for BA LoA representation and those seven articles were needed. As a review paper, more details are needed. What are the strength and lacking points of those articles? What are the author thoughts on current key points of expressing BA LoA? What is missing and what could be added in future? As a review paper, audience expectation will more in terms of the details of standard method of reporting BA LoA.
Answer: Indeed, the paper is of hybrid nature which is partly due to the aim of reviewing methodological reviews, comprising both systematic reviews and commentaries (e.g., Berthelsen-Nilsson 2006, ref. #33), and partly due to our intent to exemplify the chosen list of reporting items by Abu-Arafeh et al. (ref. #23), highlighting that efforts for a more detailed and transparent Bland-Altman analysis actually are doable within limited space. In practice, Supplemental Materials could be used more to this end.
The reviewer thought it can be better to divide this article into two separate articles. Right now, the current status of the manuscript is neither a full review article nor an original research article. One as a full review paper where more details to present. Second can be an application paper where author can utilize his data to determine the items to present BA LoA with more details of each item.
Answer: I can definitely see your point as such an approach would separate the review from the worked example; however, I think the readers of Diagnostics will benefit more from my hybrid proposal of a review, despite its limitations (see Section 4.2 for limitations of the study), combined with a hands-on worked example of which data are accessible as Suppl. Data 2.
Reviewer 2 Report
The abstract doesn’t fully explain the aim of the study, how it was done, and exactly how the reviews were compared. Was the example study used as a test for the advice articles that were found? The last paragraph of the introduction mentions consolidation of experience, but this is not really addressed. In fact, the final impression is of a “worked example”: is this the only message we can take from a systematic review of reviews?
What was the exact date range sampled from PubMed? Were any steps taken to increase the specificity of the PubMed search? Previous reports have indicated that PubMed wasn’t the best source of relevant articles, and that alternative methods such as Citation Index are more specific. The exact format of the MeSH terms used would be helpful. The terms Bland and Altman were used, but were these used with NOT (au)? PRISMA suggest: “Present full electronic search strategy for at least one database, including any limits used, such that it could be repeated.”
Page 2, l 70 “The list of article summaries that emerged from the search was screened, and potentially relevant titles were chosen for a full-text review.” How was this done, for 5,600 titles? My optimistic assessment is that it would take me about 6 hours merely to read the titles. However the author states “All papers that proposed a list of items for standardizing BA plots were included” which suggests that many of the articles had been read. A PRISMA flow diagram would be helpful here: numbers of records at each step of the search, screen, eligibility, synthesis. The PRISMA checklist asks for method of data extraction from reports (e.g., piloted forms, independently, in duplicate)
Isn’t it presumable that Criterion 16 would be covered by the ethics of authorship?
In the clinical example, perhaps a mention of the units of measurement would be helpful, with a central value to give a feel for the magnitude. Here, the mean area appears to be about 25 cm2 and the LOA is about 5 cm2.
The example paper includes a correlation assessment between measures: Bland and Altman would not approve!
I find it hard to agree completely with the statement on page 5 line 116:
“The interpretation of the BA LoA of -2.39 to 2.36 with outer 95% confidence limits -2.71 and 2.69 were post hoc judged to be sufficiently close from a clinical point of view”. This very much depends on how the data are to be used. This is a large epidemiological study of the relationship between left atrial area and other measures such as age, aortic size and renal function, and significant associations were found. However, this is because a large sample was used. For an individual patient, a single measure of atrial area is going to be sufficiently imprecise as to be of no clinical value whatsoever. As we say, “horses for courses”!
Looking at the data supplied, I presume that the first 20 rows represent the first duplicate observer measurements of 20 patients, the second 20 rows the second with another 20 patients, and so on. Here we have two sources of variance, the observer, and the patients: we need to be told how this was managed statistically.
On page 8 line 187, the author admits that a single worker is likely to have problems with bias and reproducibility: again the PRISMA guidelines are clear about how bias should be addressed. This is a substantial weakness in the method.
I am afraid that there is a history of attempts to improve statistical quality, not limited to this topic, that have failed: see, for example
Guidelines for reporting statistics in journals published by the American Physiological Society: the sequel. Curran-Everett and Benos, Adv Physiol Educ 31: 295–298, 2007
Poor statistical reporting, inadequate data presentation and spin persist despite editorial advice. Joanna Diong, Annie A. Butler, Simon C. Gandevia, Martin E. Heroux. PLoS ONE 13(8): e0202121. https://doi. org/10.1371/journal.pone.0202121
It could be sobering to discuss this persistent lack of influence of statistical advice. Another important item for discussion is the question of sample size. This is more generally used to assess “significance” that is the chance of false conclusions when a null hypothesis is posited, but as alluded to above, it all depends what the measurement is to be used for. In a single patient we may well require an entirely different precision than we might in an epidemiological study.
The paper is clearly written with only rare errors in the grammar but the style is dense and sometimes awkward. Revision using shorter sentences and less words would make it simpler to follow. As a single example, consider Page 4 lm 30-35. Here I would suggest:
A pair of observations is made from the same subject, with two different methods. The means and differences of these pairs of values for each subject are displayed in a scatter plot. The plot usually also shows a line for the estimated mean difference between the two methods (a measure of the bias between the two methods), and lines indicating the 95% confidence limits (within which approximately 95% of all population differences would lie) [1–4]. Using 95% limits of agreement assumes that the differences are normally distributed.
Sometimes, the meaning could be improved. For example
“BA LoA are sample-based estimates of the true, but unknown population limits, and Bland and Altman proposed approximate confidence intervals to assess the precision of these estimates back then [1]” would be better put as “… originally proposed approximate confidence intervals to assess the precision of these estimates”.
A single author always knows what his meaning is, and may not realise that others may not! Careful revision by a native English speaker, preferably one familiar with scientific editing, would help make the text simpler for a reader who hasn’t read the article many times.
Author Response
Dear Reviewer 2.
Thank you very much for your comprehensive and helpful comments. The works of Curran and Benos as well as Drummond and Vickers have been most illuminating. In the following, all items raised are addressed point-by-point.
Best wishes, Oke Gerke
The abstract doesn’t fully explain the aim of the study, how it was done, and exactly how the reviews were compared. Was the example study used as a test for the advice articles that were found? The last paragraph of the introduction mentions consolidation of experience, but this is not really addressed. In fact, the final impression is of a “worked example”: is this the only message we can take from a systematic review of reviews?
Answer: In the abstract, the aim of identifying the most comprehensive and appropriate list was added. In Section 2.1, more details on the review have been added. However, the reviews were simply compared with respect to the items proposed by each list. The last paragraph of the introduction has been modified accordingly. Moreover, the conclusion of the abstract was modified to highlight the purpose of the worked example, whereas the general recommendation remains to be the propagated increased use of the list suggested by Abu-Arafeh et al. (ref. #23).
What was the exact date range sampled from PubMed? Were any steps taken to increase the specificity of the PubMed search? Previous reports have indicated that PubMed wasn’t the best source of relevant articles, and that alternative methods such as Citation Index are more specific. The exact format of the MeSH terms used would be helpful. The terms Bland and Altman were used, but were these used with NOT (au)? PRISMA suggest: “Present full electronic search strategy for at least one database, including any limits used, such that it could be repeated.”
Answer: In Section 2.1, more details on the search have been added, and the new Suppl. Data 1 shows a screenshot of the conducted MEDLINE/PubMed search. In Section 4.2, the main limitations of the study with respect to the search itself have been extended.
Page 2, l 70 “The list of article summaries that emerged from the search was screened, and potentially relevant titles were chosen for a full-text review.” How was this done, for 5,600 titles? My optimistic assessment is that it would take me about 6 hours merely to read the titles. However the author states “All papers that proposed a list of items for standardizing BA plots were included” which suggests that many of the articles had been read. A PRISMA flow diagram would be helpful here: numbers of records at each step of the search, screen, eligibility, synthesis. The PRISMA checklist asks for method of data extraction from reports (e.g., piloted forms, independently, in duplicate)
Answer: Most titles have actually been obviously out of scope or an application of BA analysis. A PRISMA flow diagram has been added as Figure 1. The limitation of this single-author endeavor has been added in Section 4.2.
Isn’t it presumable that Criterion 16 would be covered by the ethics of authorship?
Answer: Yes, thank you; the description of criterion 16 has been added to this end in Section 3.1.
In the clinical example, perhaps a mention of the units of measurement would be helpful, with a central value to give a feel for the magnitude. Here, the mean area appears to be about 25 cm2 and the LOA is about 5 cm2.
Answer: Yes, thank you; mean and range of left atrium size including units have been added to Section 3.2.1. Units have also been added to Figs. 2 and 3.
The example paper includes a correlation assessment between measures: Bland and Altman would not approve!
Answer: At the time of writing this piece, the main paper (ref. #25) was under review. The main paper is, though, currently under revision and correlation assessment has been deleted.
I find it hard to agree completely with the statement on page 5 line 116:
“The interpretation of the BA LoA of -2.39 to 2.36 with outer 95% confidence limits -2.71 and 2.69 were post hoc judged to be sufficiently close from a clinical point of view”. This very much depends on how the data are to be used. This is a large epidemiological study of the relationship between left atrial area and other measures such as age, aortic size and renal function, and significant associations were found. However, this is because a large sample was used. For an individual patient, a single measure of atrial area is going to be sufficiently imprecise as to be of no clinical value whatsoever. As we say, “horses for courses”!
Answer: Acknowledged. This part of Section 3.2.1 was rephrased and kept neutral. Indeed, Bland-Altman Limits of Agreement are supposed to be valid for a wide age range, for instance, in our example. At best, one can hope to get an unselected, random sample, but I can see that reproducibility of left atrium size measurement within up to 3 cm2 is of little value.
Looking at the data supplied, I presume that the first 20 rows represent the first duplicate observer measurements of 20 patients, the second 20 rows the second with another 20 patients, and so on. Here we have two sources of variance, the observer, and the patients: we need to be told how this was managed statistically.
Answer: Yes, indeed, thank you. Supplemental Data 2 have been modified to long format and extended by a variable for rater (0: physician with n=140; 1-7: radiographers with n=20 each) and another variable for repetition (1,2 – only radiographers). In Section 3.2.3, repeatability coefficients have been added which were based on a linear mixed model. The results of 2.80 and 2.98 were contrasted with slightly underestimated BA LoA and their outer 95% confidence limits in Section 4.2 (limitations of the study).
On page 8 line 187, the author admits that a single worker is likely to have problems with bias and reproducibility: again the PRISMA guidelines are clear about how bias should be addressed. This is a substantial weakness in the method.
Answer: This main limitation has been more clearly stated in Section 4.2 now.
I am afraid that there is a history of attempts to improve statistical quality, not limited to this topic, that have failed: see, for example
Guidelines for reporting statistics in journals published by the American Physiological Society: the sequel. Curran-Everett and Benos, Adv Physiol Educ 31: 295–298, 2007
Poor statistical reporting, inadequate data presentation and spin persist despite editorial advice. Joanna Diong, Annie A. Butler, Simon C. Gandevia, Martin E. Heroux. PLoS ONE 13(8): e0202121. https://doi. org/10.1371/journal.pone.0202121
It could be sobering to discuss this persistent lack of influence of statistical advice. Another important item for discussion is the question of sample size. This is more generally used to assess “significance” that is the chance of false conclusions when a null hypothesis is posited, but as alluded to above, it all depends what the measurement is to be used for. In a single patient we may well require an entirely different precision than we might in an epidemiological study.
Answer: Yes, thank you – though the reading of these materials actually was both depressing and sobering. I was aware of the BMJ series by Bland and Altman, but not of the J Physiol series by Drummond and Vickers. The latter is very helpful for any biostatistician of a research unit of clinical physiology and nuclear medicine (like me). The discussion has been extended by Sections 4.4.1 (Sample size considerations) and 4.4.2 (Insufficient vigor of statistical advice) which both end with a personal perspective on these issues.
The paper is clearly written with only rare errors in the grammar but the style is dense and sometimes awkward. Revision using shorter sentences and less words would make it simpler to follow. As a single example, consider Page 4 lm 30-35. Here I would suggest:
A pair of observations is made from the same subject, with two different methods. The means and differences of these pairs of values for each subject are displayed in a scatter plot. The plot usually also shows a line for the estimated mean difference between the two methods (a measure of the bias between the two methods), and lines indicating the 95% confidence limits (within which approximately 95% of all population differences would lie) [1–4]. Using 95% limits of agreement assumes that the differences are normally distributed.
Sometimes, the meaning could be improved. For example
“BA LoA are sample-based estimates of the true, but unknown population limits, and Bland and Altman proposed approximate confidence intervals to assess the precision of these estimates back then [1]” would be better put as “… originally proposed approximate confidence intervals to assess the precision of these estimates”.
A single author always knows what his meaning is, and may not realise that others may not! Careful revision by a native English speaker, preferably one familiar with scientific editing, would help make the text simpler for a reader who hasn’t read the article many times.
Answer: Thank you very much for the hints. The language and grammar of the revised manuscript has been checked and corrected by Editage.
Round 2
Reviewer 1 Report
This is a resubmission. Previously reviewer raised some questions regarding the criteria of how articles were included in the study to review. Authors have answered the question and included a flow chart to explain this.
Reviewer also asked question of not reporting some calculation which have been discussed in the revised manuscript.
The main concern of the reviewer from the last time was the nature of the article that it summarized the previous articles as a review paper and then later presented newly data analysis. But as authors mentioned, the paper itself is a hybrid nature. The article aims to review the methodology and exemplify them. The authors also added some supplemental materials.
Overall, the quality of the revised manuscript has improved from the last submission.
Reviewer 2 Report
The paper had been extensively revised. The review of the search results still seems heroic - a tighter search strategy would have saved some time.